

# Teratological changes in postembryos of *Eratigena atrica* obtained by the application of alternating temperatures on spider embryos

Teresa Napiórkowska[1], Julita Templin[1] and Paweł Napiórkowski[2]

[1] Faculty of Biological and Veterinary Sciences, Department of Invertebrate Zoology and Parasitology, Nicolaus Copernicus University of Torun, Toruń, Poland
[2] Faculty of Biological Sciences, Department of Hydrobiology, Kazimierz Wielki University in Bydgoszcz, Bydgoszcz, Poland

## ABSTRACT

Spider embryonic development depends on several factors, including temperature. Under optimum thermal conditions embryogenesis proceeds undisturbed and embryo mortality is low. On the other hand, dramatic shifts in incubation temperature may cause a range of developmental defects in embryos. It has been confirmed in numerous laboratory experiments that abrupt temperature changes can be a powerful teratogenic factor. Changes in the external structure are frequently reflected in the internal anatomy, and above all, in the central nervous system. In the present teratological study, by exposing spider embryos to the temperatures of 14 °C and 32 °C, changed every 12 hours for the first 10 days of their development, we obtained 74 postembryos of *Eratigena atrica* with body deformities such as oligomely, heterosymely, schistomely, bicephaly, complex anomalies and others. We selected six spiders to describe and analyze their morphological changes. In one case, that of a spider affected by polymely (the presence of a supernumerary appendage) combined with heterosymely (the fusion of walking legs), we also focused on the structure of the central nervous system. The analysis indicated that this complex anomaly was accompanied by only one change in the central nervous system: the presence of a supernumerary neuropil. Since no fusion of walking leg neuropils was observed, it was concluded that, in this instance, there was no relationship between the fusion of legs and the structure of the central nervous system.

Corresponding author
Paweł Napiórkowski,
pnapiork@ukw.edu.pl

## INTRODUCTION

Teratology, a relatively new field of knowledge, only coming into existence in the early 19th century, studies the causes, mechanisms, and patterns of abnormal development (*Ujházy et al., 2012* and references therein). Owing to advances in genetics, toxicology, molecular biology, animal testing, and research on living organisms-environment interactions, teratology has developed significantly in recent years (*Calado & dos Anjos Pires, 2018*). Currently there are many known teratogenic factors. Their teratogenicity has been

confirmed in numerous experiments, which aid our understanding of both developmental defects and their mechanisms and normal processes occurring during embryogenesis (*Wilson, 1964*). According to the principles of teratology/developmental toxicology, toxins acting on embryos cause dysmorphogenesis when applied in a sufficient dose during a sensitive period in the development of a sensitive species. Though in vitro studies can provide reliable means to assess the potency of teratogenic/toxic substances, there is still a need to use animal models to demonstrate their embryotoxicity (*Carvan III et al., 2004*). Invertebrates seem to be particularly useful for testing the toxicity/teratogenicity of different factors, including environmental ones. These animals occupy key positions in the food chain, in aquatic and terrestrial ecosystems, and some species or groups of species are found throughout the entire habitat. They have been used for decades in toxicity tests so they have an enormous potential to help identify environmental hazards. Invertebrates have a number of characteristics that facilitate their breeding including small size, high fertility rate and short lifespan. These factors, together with low purchasing cost, ensure relatively easy and very efficient application in laboratory testing (*Lagadici & Caquet, 1998*).

Arthropods, including spiders, in which the body is divided into the prosoma and opisthosoma, are considered to be excellent models for teratological research. Various teratogenic agents applied to spider embryos may cause deformities in both tagmata. Most commonly, these defects are found on the prosoma and its appendages and are easy to detect. In addition, processing a histological specimen for examination, i.e., for an assessment of changes in the internal structure, is a straightforward task. The synanthropic spider *Eratigena atrica* (C.L. Koch) (previously *Tegenaria atrica*) from the family Agelenidae has been widely used in teratology research. Important features of this species are the relatively long breeding season in autumn/winter, high fertility rate and large embryo size. A number of experiments have been carried out to induce developmental deformities in this spider species (*Jacuński, 1969*; *Napiórkowska, Jacuński & Templin, 2010b*; *Napiórkowska, Napiórkowski & Templin, 2016a*; *Napiórkowska & Templin, 2017a*; *Napiórkowska & Templin, 2017b*; *Napiórkowska & Templin, 2018*).

Various developmental abnormalities in natural populations of terrestrial and aquatic animals have been documented so far. There is an abundant amount of information on body deformities in many groups of arthropods including crustaceans, insects, myriapods, and chelicerates (e.g., *Asiain & Márquez, 2009*; *Ćurčić et al. (1991)*; *Estrada-Peña, 2001*; *Fernandez, Gregati & Bichuette, 2011*; *Ferreira, 2011*; *Feuillassier et al., 2012*; *Kozel & Novak, 2013*; *Leśniewska et al., 2009*; *Levesque et al., 2018*). Since many of these teratogenically modified animals were collected in the natural environment, the causes of their deformities remain unknown. Numerous biological factors (including mutations of the germ or somatic cells) as well as mechanical, physical, and chemical ones can be considered as possible determinants of anomalies in arthropods (e.g., *Miličić, Pavković-Lučić & Lučić, 2013*). Deformities can also be induced in strictly controlled laboratory conditions using certain teratogenic agents, e.g., chemical reagents such as cytochalasin B, dithiothreitol, $\alpha$- lipoic acid, $NaHCO_3$, manganese, lead (*Itow & Sekiguchi, 1979*; *Itow & Sekiguchi, 1980*; *Köhler et al., 2005*; *Pinsino et al., 2010*) and colchicine (*Buczek et al., 2019*), radiation such as X-rays (*Matranga et al., 2010*), and physical parameters such as humidity (*Buczek,*

*2000*) and temperature. *Holm (1940)* was the first to put forward a hypothesis about a teratogenic effect of temperature on spiders. Later, *Juberthie (1962)* studied the impact of elevated temperature on the embryonic development of harvestmen. In subsequent years laboratory experiments confirmed the thesis that abrupt temperature changes could be a powerful teratogenic factor for *Eratigena atrica* embryos (*Jacuński, 1971*; *Jacuński, 1984*; *Jacuński & Templin, 2003*; *Napiórkowska, Jacuński & Templin, 2010a*; *Napiórkowska, Jacuński & Templin, 2010b*; *Napiórkowska, Napiórkowski & Templin, 2016a*; *Napiórkowska, Napiórkowski & Templin, 2016b*). The application of alternating temperatures (lower and higher than the optimum) during the early stages of embryonic development of *E. atrica* led to a range of deformities of the prosoma and opisthosoma (*Jacuński, 1984*). Understandably, some of these changes prevented deformed individuals from going through successive stages of postembryogenesis. With seriously impaired locomotion, they were unable to hunt, feed, moult and reproduce. Numerous anomalies, including oligomely (absence of one or more appendages), symely (fusion of appendages of the same pair), schistomely (bifurcation of appendages), heterosymely (fusion of adjacent appendages), polymely (appearance of one or more additional appendages), bicephaly (presence of two heads), and so-called complex anomalies (several anomalies occurring simultaneously), have been identified in teratogenic studies (*Jacuński & Napiórkowska, 2000*; *Jacuński, Templin & Napiórkowska, 2005*; *Napiórkowska & Templin, 2013*; *Napiórkowska, Jacuński & Templin, 2007*; *Napiórkowska, Napiórkowski & Templin, 2015*; *Napiórkowska, Templin & Napiórkowski, 2013*). Some of them (oligomely) were observed with high frequency, others were quite rare (bicephaly) (*Jacuński, Templin & Napiórkowska, 2005*; *Templin, Jacuński & Napiórkowska, 2009*). In many instances, the description of morphological defects was followed by a histological analysis of deformed spiders. Particular attention was paid to the central nervous system (*Napiórkowska, Jacuński & Templin, 2010a*; *Napiórkowska, Jacuński & Templin, 2010b*; *Napiórkowska & Templin, 2017a*; *Napiórkowska, Templin & Wołczuk, 2017*).

The structures of the digestive and nervous systems have been extensively analyzed in individuals with complex anomalies. The results indicated different internal effects depending on the anomaly (*Napiórkowska, Napiórkowski & Templin, 2015*; *Napiórkowska, Templin & Wołczuk, 2017*): morphological deformities were not always reflected in the internal anatomy. Therefore, preparation of histology slides of individuals affected by new types of complex anomalies would facilitate the classification of the defects.

In the breeding season 2017/2018 the application of alternating temperatures during early embryogenesis of *Eratigena atrica* provided us with new, interesting cases in the teratogenic material. Although this method has been used in teratology research on this spider species for years, it can still produce unpredictable results. These new, random anomalies are worth discussing in detail. Our study also emphasizes the power of temperature as a teratogenic agent, whose application in the laboratory may cause such extensive changes in spider anatomy and morphology that affected individuals are unable to express normal behaviour or develop a reproductive strategy. Therefore, the aim of the study was to show the diversity of anomalies in terms of morphological changes as well as, in one spider,

anatomical changes. In the latter case, it was hypothesized that morphological changes were reflected in the structure of the central nervous system.

## MATERIAL AND METHODS

The teratological experiment involved embryos of *Eratigena atrica* (CL Koch, 1843). 24 sexually mature females and 17 males were collected in early autumn near the towns of Chełmża and Toruń (Poland) and transported to the laboratory, where each spider was put into a separate glass container with a capacity of 250 cm$^3$. Spiders were kept in a dark room at the temperature of 21–23 °C and relative humidity (RH) of 70%. Three weeks after the culture was established, males were introduced into containers with females ready for insemination. First egg sacs were laid after several weeks. Embryos were removed from each egg sac, counted, and divided equally into two groups: experimental and control. The control group was kept at the temperature of 22 °C and 70% RH, while the experimental group was exposed to temperatures of 14 °C and 32 °C (70% RH) applied alternately every 12 h. The procedure continued for ten days, until first metameres of the prosoma appeared on the germ band and the leg formation process began (embryo development was observed in paraffin oil; the chorion becomes transparent in paraffin oil). Subsequently, all experimental embryos were incubated under the same conditions as the control ones. After hatching, all control and experimental postembryos (by *Downes, 1987*) were examined for developmental deformities on the prosoma and opisthosoma. Deformed individuals were photographed using a light microscope (Axcio Lab A1). Images were recorded using a digital camera (Axiocam 105 color, Carl Zeiss) and a computer system running Zen software (Version 2.3, blue edition). One individual was subjected to histological analysis, in which 7-$\mu$m-thick paraffin sections were stained with Mayer hematoxylin and eosin (Mayer's haemalum technique). The same method was used to make histological preparations of one individual from the control group.

## RESULTS

In the breeding season 2017/18 we obtained approximately 6,000 embryos, half of which constituted the control group. Individuals from this group were not affected by any developmental defects and the mortality rate was low (8%). They had six pairs of normally developed prosomal appendages and histological analysis of one individual showed no aberrations in the central nervous system (Fig. 1).

In contrast, in the experimental group the mortality rate of embryos was much higher (37%). This group contained several postembryos with teratogenic changes on the prosoma or opisthosoma. In total, 74 out of 1,900 postembryos were affected by one of the following anomalies within the prosoma: oligomely, heterosymely, schistomely, bicephaly, complex anomaly and others. In the latter group were postembryos with considerably shorter appendages of the prosoma, protuberances of different size and shape or anomalies in the spinning apparatus (Table 1). Several interesting cases selected for analysis are presented in Fig. 2.

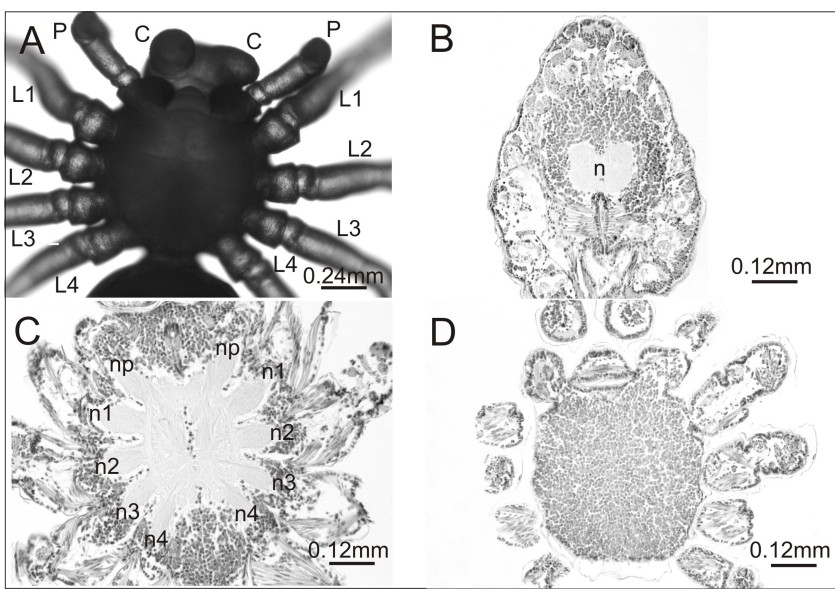

**Figure 1** *Eratigena atrica* **postembryo from control group.** (A) Ventral view: (C) chelicerae; P, pedipalps, L1–L4, walking legs; (B–D) horizontal sections through the prosoma, brain (B) and ventral nerve cord (C, D): n, neuropil; n1–n4, neuropils of walking legs; np, neuropils of pedipalps.

**Table 1** Types and frequency of anomalies on the prosoma in *Eratigena atrica* postembryos.

| Kind of anomaly | Number of individuals | % |
| --- | --- | --- |
| Oligomely | 39 | 52.70 |
| Heterosymely | 3 | 4.06 |
| Schistomely | 1 | 1.35 |
| Bicephaly | 3 | 4.06 |
| Complex anomalies | 11 | 14.86 |
| Others | 17 | 22.97 |
| Total | 74 | 100.00 |

The spider in Fig. 2A was affected by bilateral oligomely and, apart from one pair of pedipalps (P), had only three pairs of walking legs (L1–L3). Additionally, it had a truncheon-shaped protuberance (A) in place of the right chelicera (ventral view). A similar protuberance (A) developed in place of one pedipalp in the spider presented in Fig. 2B. The remaining appendages on the prosoma, i.e., chelicerae (C), left pedipalp (P) (dorsal view) and walking legs (L1–L4) were well-developed and had the correct size and segmentation. In the spider in Fig. 2C a complex anomaly was observed. The specimen had one chelicera (C), one pedipalp (P) and only two walking legs (L1 and L2) on the right side of the prosoma (ventral view). The left side of the prosoma was significantly changed. Behind the chelicera (C) there was a schistomelic pedipalp (P), with one free end much shorter and deformed. Behind the pedipalp there was also a short protuberance (A), widened in the middle, and only two walking legs (L1 and L2). A complex anomaly was also recognized in the spider in Fig. 2D. This individual was affected by oligomely of walking legs on

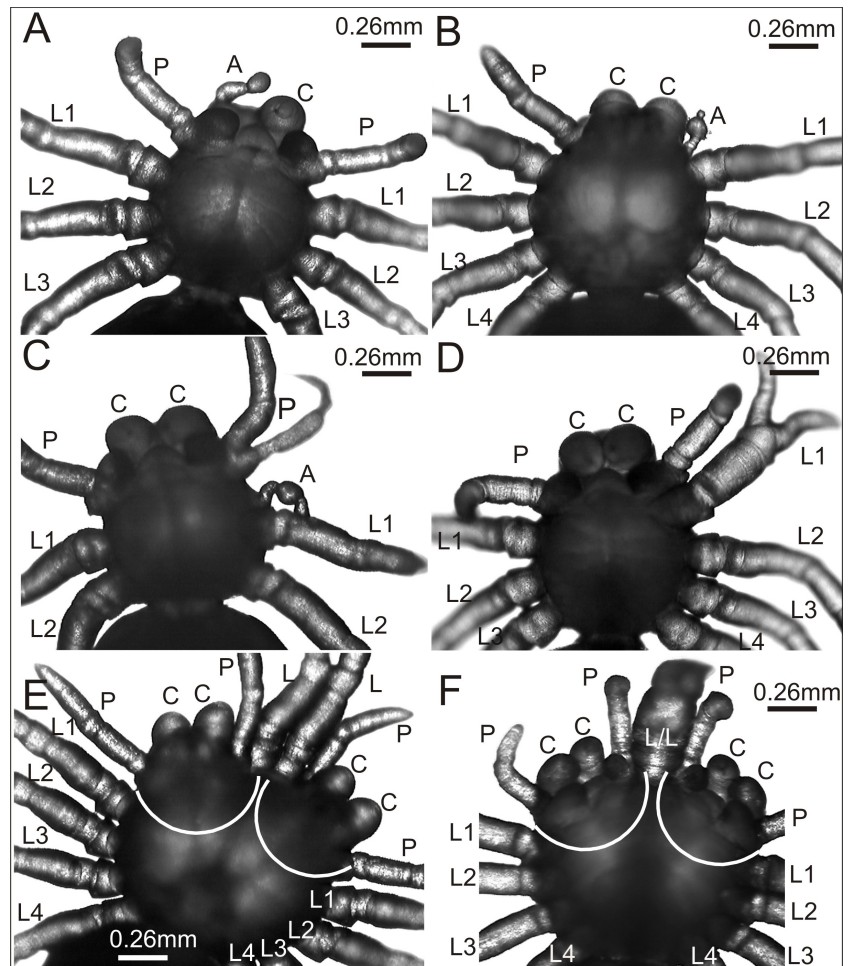

**Figure 2** *Eratigena atrica* **postembryos with teratogenic changes.** (A) Ventral view: postembryo with bilateral oligomely and a protuberance in place of the right chelicera; (B) dorsal view: postembryo with a protuberance in place of the right pedipalp; (C ) ventral view: postembryo with oligomely of the walking legs, schistomely of the left pedipalp, and a protuberance between the pedipalp and walking leg; (D) ventral view: postembryo with oligomely of the walking legs on the right side of the prosoma and schistomely of the first left walking leg; (E) dorsal view: bicephalous postembryo with additional, well-developed walking legs between two heads; (F) ventral view: bicephalous postembryo with additional, partially fused walking legs between two heads; A, protuberance; C, chelicera; L, L/L, L1–L4, walking legs; P, pedipalp; white lines indicate the heads of bicephalous postembryos.

the right side of the prosoma (ventral view), because behind the fully formed chelicera (C) and pedipalp (P) there were only three walking legs (L1–L3). On the left side of the prosoma the first walking leg (L1) was schistomelic. The bifurcation started at the tibia. The non-bifurcated part of the leg was thicker than usual, with distinct segmentation. The two free ends, which extended in opposite directions, were also distinctly segmented. Posterior to the schistomelic leg were three well-developed walking legs (L2–L4). Figs. 2E and 2F show two different bicephalous specimens. One (Fig. 2E) had two equivalent heads with a double set of chelicerae (C) and pedipalps (P). Between the heads were two fully

formed, separate walking legs (L). The other specimen (Fig. 2F) had two complete heads with chelicerae (C), pedipalps (P) and two walking legs (L/L) in between, fused from coxa to the patella. Both individuals also had a standard set of walking legs (L1–L4).

A histological analysis was performed on one individual affected by an anomaly that had not been previously recorded. The case seemed interesting both in terms of morphology and internal anatomy. As can be seen in Fig. 3A the spider had a complex anomaly, i.e., polymely and heterosymely of the walking legs on the right side of the prosoma (dorsal view). Behind a well-formed, six-segmented pedipalp (P) was a very thick, significantly deformed appendage (a) with two free ends (L1; L2A/2B). Based on its unique appearance, it was assumed that it consisted of three walking legs, which would mean that five walking legs developed on this side of the prosoma (polymely). The last two legs (L3; L4) were well-developed with seven segments. The first three were heterosymelic. Two of them (L2A/2B) were fused over their entire length (total heterosymely). In addition, they were fused with the first leg (L1) along the coxae, trochanters, femurs, and patellas (partial heterosymely), which explains the presence of the two free distal ends: L1 and L2A/2B. The assumption was that the end L2A/2B was much thicker because it was composed of the last three segments of the completely fused legs. The segmentation of this end was very indistinct, but its length was the same as that of the end of L1, which consisted of three segments. On the opposite side of the prosoma there was a set of properly formed appendages.

To verify our assumptions we prepared histology slides of the central nervous system of the investigated spider. There were no structural changes in the brain (Fig. 3B). However, in the ventral nerve cord (subesophageal ganglia) the number of leg neuropils was higher. Figure 3C shows the neuropils (n) of the ventral nerve cord in its middle part, with four separate walking leg neuropils (n1, n2A, n3, n4) in addition to the neuropil of the pedipalp (np) on the deformed right side. The last two on this side of the prosoma (n3, n4) were the neuropils of the well-formed walking legs L3 and L4: the first two (n1, n2A), those of the heterosymelic legs. Based on the location, n1 was assumed to be the neuropil of the leg whose distal end was marked as L1 and n2A was neuropil for one of the totally fused L2A/2B (end L2A/2B). The ventral nerve cord contained one abnormal additional neuropil, displaced ventrally (n2B) (Fig. 3D). Histological analysis indicated that it belonged to the second leg of the fused complex, whose end was marked as L2A/2B. No fusion of the leg neuropils was observed.

## DISCUSSION

For the present study, 1,900 postembryos that left their eggshells after exposure to the teratogenic agent (alternating sub- and supra-optimal temperatures) were examined for developmental deformities. 74 individuals, i.e., 3.9%, had body defects. Assuming that thermal shocks applied during spider embryogenesis are a potent teratogen, the number seems low. There may be multiple possible reasons of such a low frequency of deformities.

Firstly, it may be connected with expression of heat shock protein (Hsp) genes. In general, Hsp genes are expressed at low levels under normal growing conditions, but their

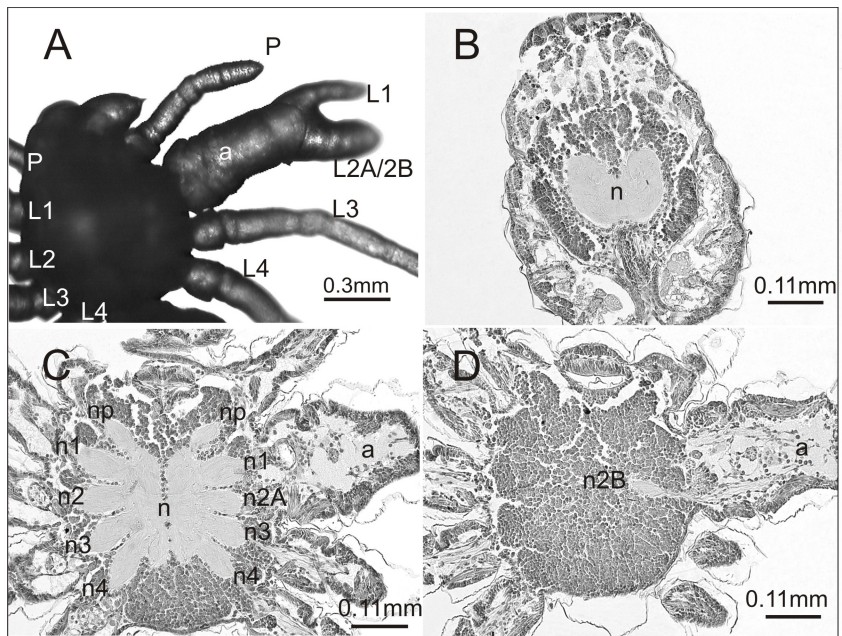

**Figure 3** *Eratigena atrica* **postembryo with complex anomaly.** (A) Dorsal view: a, deformed appendage; L1, L2A/2B, free ends of fused walking legs; P, pedipalps; L1–L4, walking legs; (B–D) horizontal sections through the prosoma, brain (B) and ventral nerve cord (C, D) (right side abnormal, left side normal): a, fused part of the legs; n1, n2A (C) and n2B (D), neuropils of heterosymelic legs; n1–n4, neuropils of the well-formed walking legs; np, neuropils of pedipalps; n, neuropil.

expression increases considerably in response to different environmental stressors such as heat, desiccation or heavy metals. Hsp genes represent a subset of a larger group of genes coding for molecular chaperones. Chaperones can assist in the efficient folding of nascent peptides, also acting when denatured proteins accumulate in cells, preventing them from irreversible aggregation and misfolding (*Martínez-Paz et al., 2014* and references therein). Secondly, effective repair processes at every stage of embryo development may eliminate errors that occur during morphogenesis. Finally, spiders, which are ectothermic animals, must be relatively resistant to abrupt temperature changes. This would also apply to spider embryos, although their mortality was relatively high (37%).

Oligomelic individuals were the most numerous in the teratological material (over 50%), which coincides with the results of previous studies (*Jacuński, 1984*; *Napiórkowska, Napiórkowski & Templin, 2016b*). A relatively large group of postembryos (23%) had deformities classified as 'Others' in Table 1, followed by postembryos with so-called complex anomalies (15%). Since many deformities obtained by the application of temperature changes during embryonic development of *Eratigena atrica* have already been described (e.g., *Napiórkowska, Napiórkowski & Templin, 2015*; *Napiórkowska, Napiórkowski & Templin, 2016a*), we focused on those that were encountered for the first time. Based on the previous observations it can be predicted that new surprising changes may occur when developmental processes are disturbed by temperature shocks. Every year novel body deformities are registered in teratological experiments.

In one case we analyzed not only deformities of the walking legs on the right side of the prosoma but also the structure of the central nervous system. The nature of this malformation suggests two processes: the formation of an additional leg (polymely) and the fusion of three walking legs (heterosymely). Only the polymely was reflected in the central nervous system: an additional walking leg neuropil was found in the ventral nerve cord, but neuropil fusion was not observed. Therefore, on the right side of the ventral nerve cord there were five walking leg neuropils and one of them was shifted to the ventral side. According to *Jacuński (1984)*, an additional leg, not developed during normal ontogenesis, is associated with the appearance of an additional half of a metamere (and thus of a neuromere) on the germ band. This suggests that all polymelic legs should have their ganglia, as has been observed in numerous studies (e.g., *Napiórkowska, Napiórkowski & Templin, 2015*). However, two different scenarios have been observed in instances of leg polymely: (1) an increased number of ganglia and their fusion, despite the absence of fused legs (*Napiórkowska, Napiórkowski & Templin, 2015*; *Napiórkowska, Templin & Wołczuk, 2017*), (2) an increased number of ganglia and no leg or ganglia fusions (*Jacuński et al., 2002*; *Napiórkowska, Jacuński & Templin, 2006*).

Based only on the morphology of the spider in Fig. 3, another possibility could be considered, namely that the fused walking legs consist of only two legs (despite relatively big difference in thickness of the two free ends). In teratological studies, leg deformities consisting of significant thickening, narrowing or curving, which are frequently observed in postembryos, disappear after several molts (personal observation). Therefore, the thicker end (L2A/2B) could, in theory, belong to one leg. However, histological analysis indicated the presence of an additional, polymelic leg, whose neuropil (n2B) was shifted to the ventral side. Additionally, intervening sections did not reveal continuity between n2A and n2B. For that reason it can be concluded that neuropil n2B is not part of a distorted n2A but of the additional leg. However, even if such a continuity had been discovered, not only complete heterosymely of the legs (L2A/2B), but also of the corresponding neuropiles (n2A and n2B) could have been a possible diagnosis.

The spatial location of the ganglia is another issue. In the majority of cases the ganglia (neuropils), including the supernumerary ones, were located in one plane (*Napiórkowska, Templin & Wołczuk, 2017*). In several individuals the ganglia were shifted to the dorsal or ventral side (*Napiórkowska, Napiórkowski & Templin, 2015*). It is therefore important to understand the causes of these shifts. First, they may be induced by changes in the genes responsible for the formation of the anterior-posterior and dorsal-ventral axes, which determine the location of all internal organs and structures. Based on latest reports (e.g., *Damen et al., 1998*; *Khadjeh et al., 2012*; *Pechmann et al., 2009*; *Schwager et al., 2009*; *Telford & Thomas, 1998*) we suppose that these shifts resulted from changes in *Hox* gene expression patterns. *Hox* genes are able to alter the arthropods plan, as well as, determine the presence or absence of legs in different parts of the body (e.g., *Antennapedia* and *Distal-less* genes). Another explanation could be space limitation: since the size and symmetry of the prosoma does not change (despite the presence of an additional leg and additional half of a neuromere) an additional ganglion has to be "pushed", ventrally or dorsally in order to fit into a limited volume of the prosoma.

In the investigated spider the heterosymely of walking legs was not associated with the fusion of their ganglia (neuropils), although it seems logical that it could have been reflected in the central nervous system. This would have suggested a certain hierarchy between segmental structures, with the fusion of ganglia leading to the fusion of the corresponding legs. Such a situation was observed in bicephalous *E. atrica* whose chelicerae and pedipalps were completely fused (*Napiórkowska et al., 2016*). However, in the vast majority of cases, heterosymely has not been accompanied by the fusion of leg ganglia (*Napiórkowska, Napiórkowski & Templin, 2015*; *Napiórkowska, Templin & Napiórkowski, 2013*). This indicates that the fusion of walking legs may result from the fusion of the developing leg buds, caused by the exposure of an early embryo to thermal shock. In other words, thermal shock may affect the developing leg buds, but not necessarily leg ganglia. Temperature changes applied during embryo incubation may or may not affect various elements of serial structures, such as ganglia or legs. The consequences depend on the intensity of the thermal shock and time of its application. From this point of view, an application of a teratogen in early stages of embryogenesis may cause more profound changes than its later application and a range of effects may be expected. Moreover, since morphological defects are not always reflected in the central nervous system (an example of which is the investigated *E. atrica*), teratological studies should not be limited to deformity descriptions but should also focus on internal anatomical examination, including that of the central nervous system. This type of research has already been conducted on spiders and other arthropods (*Harzsch, Benton & Beltz, 2000*; *Jacuński, Templin & Napiórkowska, 2005*; *Scholtz, Ng & Moore, 2014*).

A certain analogy can be seen between the investigated *E. atrica* and a deformed pycnogonid *Pycnogonum litorale* described by *Scholtz & Brenneis (2016)*. In the latter, an extensive deformity resulted from a mechanical, unintentional injury to the region between the second and third walking leg. After several months the sea spider developed an extra leg on the right side of the prosoma partially fused with other legs. In this case, the supernumerary leg did not have an associated ganglion although it did, like the other legs, contain a midgut diverticulum and a branch of ovary. *Scholtz & Brenneis (2016)* explained the anomaly using the "boundary model" proposed by Meinhardt in the 1980s and supported later by molecular data. This model hypothesized the division of each body segment into at least three cellular compartments, designated S, A and P, along the antero-posterior axis (*Meinhardt, 1986*). Two of these were known from *Drosophila* research, with each segment comprised of transverse cell populations with an anterior or a posterior fate, the A and P compartments, respectively, which lie strictly separated but adjacent to each other (*Martinez-Arias & Lawrence, 1985*). Meinhardt' model further hypothesized that, perpendicular to each A-P border, there is a longitudinal boundary separating dorsal (D) and ventral (V) cells on either lateral side of the embryo (*Meinhardt, 1986*). At intersections between A-P and D-V borders, the formation of limb buds is initiated. If the cells of an S compartment are removed, the P cells of an anterior segment form a contact zone with the A cells of the more posterior segment and an additional leg is formed. This model could also be used to explain the formation of an additional leg in the investigated *E. atrica*. Furthermore, molecular analysis might help explain the mechanisms

of morphological defects in this spider. Many researchers, including *Khadjeh et al. (2012)* and *Pechmann et al. (2011)* have successfully conducted such studies on spiders.

All developmental defects, both those that are caused by some complex regenerative processes and those that are caused by teratogenic factors (e.g., alternating temperatures applied during early embryogenesis), can contribute to a better understanding of spider phylogenesis and development. For example, the abnormal development of a short appendage on the petiolus (pedicel) in a postembryo of *E. atrica*, as observed by *Jacuński & Templin (1991)*, has been interpreted as an atavistic feature which might facilitate determining how spiders are related to other groups of arthropods. Obviously, this issue would require in-depth research. In addition, developmental defects indicate morphological capabilities and developmental potential of organisms, not displayed under normal conditions.

Teratological experiments based on temperature seem justified in view of current weather anomalies. Sudden temperature changes observed nowadays can affect embryos, causing damage that may not be subject to spontaneous repair processes. As a consequence, a higher number of deformed animals may be found in the natural environment. Observation of spiders, commonly found near human settlements, can provide abundant evidence of adverse environmental impacts.

In conclusion, our results suggest that temperature changes during embryonic development of animals can cause various deformities in their body structure. Our findings provide additional evidence that morphological defects are not always reflected in the central nervous system (an example of which is the investigated *E. atrica*). Therefore, teratological studies should not be limited to describing external features of deformed individuals, but should also involve analyzing their internal organs, including the CNS.

### Funding

This work was supported by the Faculty of Biological and Veterinary Sciences of the Nicolaus Copernicus University in Toruń [statutory fund research] and Faculty of Biological Sciences of the Kazimierz Wielki University in Bydgoszcz (Poland). The funders had no role in study design, data collection and analysis, decision to publish, or preparation of the manuscript.

### Grant Disclosures

The following grant information was disclosed by the authors:
Faculty of Biological and Veterinary Sciences of the Nicolaus Copernicus University in Toruń.
Faculty of Biological Sciences of the Kazimierz Wielki University in Bydgoszcz (Poland).

### Competing Interests

The authors declare there are no competing interests.

## Author Contributions

- Teresa Napiórkowska conceived and designed the experiments, performed the experiments, analyzed the data, authored or reviewed drafts of the paper, and approved the final draft.
- Julita Templin conceived and designed the experiments, analyzed the data, prepared figures and/or tables, and approved the final draft.
- PawełNapiórkowski performed the experiments, authored or reviewed drafts of the paper, preparation of the manuscript for revision, and approved the final draft.

## Data Availability

The raw data is available in the table and figures.

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
