# Peer review of "Teratological changes in postembryos of Eratigena atrica obtained by the application of alternating temperatures on spider embryos"

_PeerJ, doi:10.7717/peerj.11457_

## Round 0.1 · original submission · Major Revisions

Dear Dr. Napiórkowska and colleagues:

Thanks for submitting your manuscript to PeerJ. I have now received three independent reviews of your work, and as you will see, the reviewers raised some concerns about the research. Despite this, these reviewers are optimistic about your work and the potential impact it will have on research studying the effect of extreme temperatures on embryogenesis. Thus, I encourage you to revise your manuscript, accordingly, taking into account all of the concerns raised by both reviewers.

NOTE: you draw attention to the species in the title but do not define it in the Abstract. I understand the findings apply to spiders in general but have applications to overall embryology, but the study organism needs to be adequately defined in the Abstract.

While the concerns of the reviewers are relatively minor, this is a major revision to ensure that the original reviewers have a chance to evaluate your responses to their concerns. There are not too many suggestions; thus, it should not take much effort to address these concerns to greatly improve your manuscript.

Please update the figures with wildtype control images.

Please note that reviewer 2 has included a marked-up version of your manuscript.

I look forward to seeing your revision, and thanks again for submitting your work to PeerJ.

Good luck with your revision,

-joe

Reviewer 1 ·

Basic reporting

No comment

Experimental design

Overall, the manuscript is written very well. The only major issue I found was the complete absence of imaging techniques in the methods section. The experimental design seems very sound, the rest of the methods were detailed, but I would recommend that full details of the preparation and imaging techniques, including types/brands of microscope be included in the methods.

Validity of the findings

This is a very interesting study and would be of interest to the wider scientific community.
I believe this study on E. atrica will provide important reference material for this particular species is widespread throughout Europe and one of the most abundant synanthropic species, ideal to compare against for studies such as climate change, or new alien, and potentially invasive species arriving in Europe, particularly those from warmer climates.

Additional comments

Line 74: I would avoid the term "It is a known fact that" and begin the sentence with "In the natural environment, spider....."

·

Basic reporting

Most of my specific comments are on an attached annotated copy of the manuscript.

One suggestion not included on that copy concerns the designations of walking legs and neuropils used in Figure 2. The symmetry evident in the subesophageal ganglia (ventral nerve cord) in Fig. 2C indicates that the left n4/L4 is part of the same metamere as the right n5/L5. Likewise, the left n3/L3 appears to be from the same metamere as the right n4/L4. If the authors agree about these correspondences, then I suggest that the structures currently designated n2/L2 and n3/L3 in Fig. 2 (and discussed in the text with these designations) be re-designated as n2A/L2A and n2B/L2B, respectively. In so doing, the right n4/L4 and n5/L5 would become n3/L3 and n4/L4, respectively, thus providing them with the same designation as their metamere counterparts on the left side of the prosoma. This will emphasize the common metameric origin of these structures.

Experimental design

no comment

Validity of the findings

The authors are well positioned to interpret the histological results, having done this kind of analysis for many years. I would just ask them to consider, and discuss in the paper, how confident they are that their interpretation of polymely in the Fig. 2 individual is correct. The small neuropil evident in Fig. 2D does demonstrate that, at the very least, there is a distortion to n2 not present in any other neuropils, but whether this constitutes a separate neuropil, reflecting a supernumerary leg fused with the first two walking legs, is something I am not yet entirely convinced of based on what is presented. But the authors may be in possession of more data (e.g., in the form of additional sections between those shown in Figs. 2C and 2D) that gives them confidence that the neuropils they designate n2 and n3 are indeed distinct and not fused. For example, if intervening sections reveal little or no continuity between n2 and n3, then I would agree they are separate and unfused neuropils, making a solid case for a supernumerary leg being part of the fused assemblage. If the authors do have additional observations that lend greater certainty to their claim that polymely exists in this individual and that fusion of legs is not reflected by fusion in the CNS (including between what they designate n2 and n3), then it would be helpful if they could describe these observations. If they do not have such supporting observations, then they might consider raising the possibility in their Discussion that the fused limb contains only two legs (despite the difference in thickness of the two free ends on the fused assemblage) and the “n3” neuropil is part of a distorted n2.

Additional comments

I enjoyed reading your paper and seeing the beautiful histology and was especially interested to read about the "boundary model" as it potentially relates to the Fig. 2 specimen.

As mentioned above, most of my specific suggestions are on an annotated copy of the manuscript, entered using the Track Changes feature in Microsoft Word. Please do take the time to fully consider them since that is where I invested most of my effort. Since PeerJ only allows a PDF to be uploaded, feel free to contact me ([email protected]) if you wish me to send my annotated version of the manuscript as a Word document, where it is easier to accept any suggestions you agree with.

·

Basic reporting

For the most part clear and unambiguous, professional English was used throughout the text. A few places of clarification for the reader would be helpful for someone not in the field. Perhaps define teratology in the first sentence for readers that aren’t familiar with the term. Terminology defining the nature of each defect is well described in the introduction, but this may also be useful information to repeat in table 1 for readers that are not familiar with the terms. Line 106 wording “ a new interesting cases” should be reworded to remove the a or the s from cases.

Authors mention in line 74 that “…spiders are exposed to a range of factors which can affect their development causing various morphological deformities.” Could the author list some examples perhaps in parentheses of these factors and provide a reference to these factors (specifically ones other than temperature)? Additionally, has this temperature effect of teratogenicity been seen in other spiders, arthropods, or other classes of invertebrates?

In line 95 it is mentioned that histological analyses of the nervous system were performed in the authors’ previous studies. It is mentioned then in the next paragraph that various abnormalities were observed depending on the defect. It would benefit the reader to know the nature of these defects in the introduction to support what the authors will be looking for in the current study.

The article is professionally and conventionally structured.

Raw data is shared, and no statistical analyses were performed or needed.

Figures are clear. However, it would be of interest to readers to add a few things to the figures and table for ease of interpretation. For the table, definitions for the abnormalities could be included in a column. Perhaps additional columns, showing percentages of these abnormalities in the authors previous studies could also be useful for comparison. Arrows pointing to the various abnormalities in the figures would be useful to the reader. While some are obvious, absent or rudimentary limbs take longer to pick out. In the bicephalous larva images a dotted line demarcating where the two heads meet would be useful. For the final figure, it would be useful to have a diagram of the normal layout of the neuropils and a wildtype to compare to (more on this later).

Experimental design

This study represents original primary research within the Aims and Scope of PeerJ.
This is an exploratory study with a clearly defined research question - What are the physical teratogenic effects of extreme temperature treatments on embryogenesis if applied at early stages?

While the question is clear and methods appropriate, relevance and the gap of knowledge this research fills are not clearly stated. What relevance does this have to the natural underpinnings of developmental biology/teratology, particularly if the results are not consistent and these abnormalities seem to occur at random (new abnormalities in this study not seen in many other studies with the same treatment)? This could be easily rectified with a clear statement of relevance in the introduction. There is some mention of this in the discussion. In line 285, the authors state that developmental defects, can “contribute to a better understanding of developmental mechanisms in invertebrates.” The authors argument would be strengthened if they could provide the next steps in using the information provided by their study to understand developmental processes.

One question I had reading the introduction was whether the temperatures used in the lab studies were ones that might be experienced in the natural environment.
Majority of the methods are described with sufficient detail & information to replicate. Authors should clarify whether one cocoon from each of the 24 females was used. One might expect that the genetic background of some cocoons might predispose them to being susceptible to temperature shifts. This is controlled for in splitting the embryos equally into treatment groups. Perhaps in the discussion, authors could address the possibility that embryos having abnormalities came from the same breeding pair. A cool future study could be to keep the embryos separated by cocoon to trace genetic susceptibility. Are those individuals with complex anomalies (more than one of the anomalies listed) double counted? For those with complex anomalies. Are there any combinations that occur together with more frequency?

Figures represent individuals with complex abnormalities. Are the defects as severe in individuals with single abnormalities? Perhaps the authors could include an additional figure with a representative image for each of the single abnormalities.

Of top priority, normal wildtype spider should be shown for comparison in all figures including sectioned images.

Validity of the findings

All underlying data have been provided. Statistical analysis is not needed but as described earlier some additional numbers would be useful for comparison (types of abnormality combinations in complex abnormalities, percentages from previous studies, etc.)

The study is well controlled by splitting embryos from each egg sac equally into treatment groups. However as previously mentioned, a control wildtype spider should be shown for comparison in all figures.

One of the most well studied areas in spider developmental biology is body segmentation and leg development. While it is mentioned in line 234, that temperature shifts could induce changes in the genes responsible for these processes, it is unclear how. Authors are encouraged to explore this literature in the discussion section to propose some gene families that may be altered (such as hox genes) and the effects that temperature may have on protein function. The space limitation argument in line 238 doesn’t make sense as an alternative to the segmentation gene idea, but seems more of an explanation of how these defects in segmentation and patterning manifest in the various defects seen. Damage to a particular cell type might be a better alternative to specific protein malfunctions and would be further supported by the boundary model that is described in detail.

Places whether the authors provide speculation is clear and conclusions are not overstated.

Additional comments

The manuscript entitled “Teratological Changes in the Eratigena atricla larvae obtained as a result of the application of alternating temperatures on spider embryos.” is an interesting continuation of the researchers work looking at the effect of extreme temperatures on embryogenesis. This study is scientifically sound, however addressing some minor points could strengthen the impact of the paper. I am suggesting that it be accepted upon minor revisions. Of highest priority is updating the figures with wildtype control images.

---

## Round 0.2 · Minor Revisions

Dear Dr. Napiórkowska and colleagues:

Thanks for revising your manuscript. The reviewers are mostly satisfied with your revision (as am I). Great! However, there are a few remaining concerns to address (per reviewer 2).

Please address these ASAP so we may move towards acceptance of your work.

Best,

-joe

Reviewer 1 ·

Basic reporting

No comment

Experimental design

No comment

Validity of the findings

No comment

Additional comments

Congratulations on this work, it is an excellent research paper.

·

Basic reporting

This is a review of a revision prepared by the authors. The authors have done a conscientious job taking the comments and suggestions of the reviewers into account in preparing their revised manuscript.

As with the original manuscript, most of my specific recommended changes are included on an annotated version of the revision that I submit with this review. These are all minor with one exception: in three instances there is a discrepancy between the view (ventral or dorsal) stated for a figure in the text versus what is stated in the figure legend (one says dorsal, the other says ventral). I would just ask the authors to review these carefully and be certain that text and figure legend are in agreement, and that both of them agree with what is seen in the actual photomicrographs.

Experimental design

No comment

Validity of the findings

An earlier concern regarding one aspect of the interpretation of the histologically-examined postembryo has been addressed very satisfactorily in the revised manuscript.

Additional comments

I have no further reservations regarding the publication of this paper. I would only ask the authors to consider the suggestions I give in the accompanying annotated manuscript and accept those they agree with.

·

Basic reporting

No comment.

Experimental design

No comment.

Validity of the findings

No comment.

Additional comments

THank you for taking the comments into account for your publication. I think that adding additional discussion on molecular basis of the abnormalities and including the wildtype spider for comparison really strengthened the paper.

---

## Round 0.3 · accepted · Accept

Dear Dr. Napiórkowska and colleagues:

Thanks for revising your manuscript based on the concerns raised by the reviewer. I now believe that your manuscript is suitable for publication. Congratulations! I look forward to seeing this work in print, and I anticipate it being an important resource for groups studying spider embryology and the effect of extreme temperatures on embryogenesis. Thanks again for choosing PeerJ to publish such important work.

Best,

-joe